# Compact task representations as a normative model for higher-order brain activity

**Severin Berger**
Champalimaud Centre for the Unknown
Lisbon, Portugal
severin.berger@neuro.fchampalimaud.org

**Christian K. Machens**
Champalimaud Centre for the Unknown
Lisbon, Portugal
christian.machens@neuro.fchampalimaud.org

## Abstract

Higher-order brain areas such as the frontal cortices are considered essential for the flexible solution of tasks. However, the precise computational role of these areas is still debated. Indeed, even for the simplest of tasks, we cannot really explain how the measured brain activity, which evolves over time in complicated ways, relates to the task structure. Here, we follow a normative approach, based on integrating the principle of efficient coding with the framework of Markov decision processes (MDP). More specifically, we focus on MDPs whose state is based on action-observation histories, and we show how to compress the state space such that unnecessary redundancy is eliminated, while task-relevant information is preserved. We show that the efficiency of a state space representation depends on the (long-term) behavioural goal of the agent, and we distinguish between model-based and habitual agents. We apply our approach to simple tasks that require short-term memory, and we show that the efficient state space representations reproduce the key dynamical features of recorded neural activity in frontal areas (such as ramping, sequentiality, persistence). If we additionally assume that neural systems are subject to cost-accuracy tradeoffs, we find a surprising match to neural data on a population level.

## 1 Introduction

Arguably one of the most striking differences between biological and artificial agents is the ease with which the former navigate and control complex environments [1]. Core functions enabling such behaviours, including working memory and planning, are typically attributed to higher-order brain areas such as the prefrontal cortex (PFC) [2, 3], and exactly these functions are thought to be lacking in today's machine learning systems [4]. Yet, it remains unclear how higher-order brain areas generate these complex behaviours, or even the simple behaviours that are often studied experimentally in rodents and primates. Specifically, both behavioural strategies and neural activities depend in complex ways on the task at hand, and these dependencies have so far evaded a satisfactory or intuitive explanation [5]. For example, in tasks that require animals to remember some information, neurons are sometimes persistently active [6, 7, 8, 9], while at other times they are sequentially active [10, 11]. Indeed, subtle changes in the timing of a task can lead to a sudden shift from one to the other [9], but the causes behind these activity shifts have remained unclear.

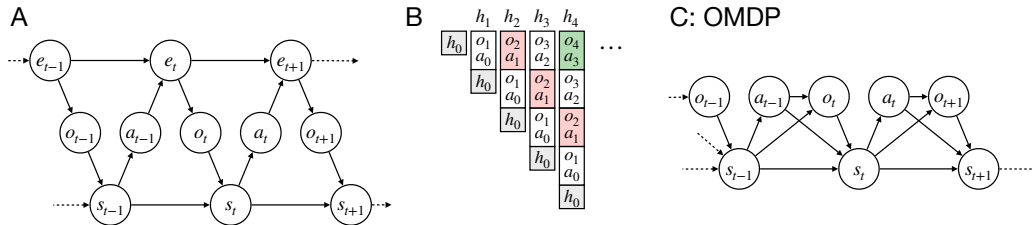

Figure 1: A: The agent-environment loop. The environment, $e$, emits observations, $o$, which include rewards. Based on its internal (belief) state, $s$, the agent chooses actions, $a$, that affect the environment. B: The most information the agent can have about the environment is to remember all past observations and actions, i.e., the history, $h$. C: Dependency graph for the OMDP.

Currently, these activity patterns are mostly studied from a mechanistic, network perspective. For instance, sequential activity has sometimes been identified with feedforward dynamics, and persistent activity with recurrent or attractor dynamics [12, 13]. More generally, task-related neural activity has been modeled by training recurrent neural networks (RNNs) to perform the same task as an animal [14, 9, 15, 16, 17, 18]. Surprisingly, RNNs can mimic recorded neurons quite well, if the task is phrased the right way, and if learning is properly regularized to avoid overfitting [19]. However, RNNs are generally difficult to interpret and analyse, although some progress has been made in this direction [20]. More importantly, training a RNN does not clarify why a particular solution is a good solution, or, indeed, if it is a good solution at all.

Here we take a step back and first define what determines a good solution. Our goal is to develop a normative approach to explain higher-order brain activities. Our starting point is the efficient coding hypothesis, which states that neural circuits should eliminate all redundant or irrelevant information [21, 22, 23]. We then merge the concept of an efficient representation with the formalism of reinforcement learning (RL) and Markov Decision Processes (MDPs). As most realistic tasks are only partially observable, we first endow the underlying MDPs with a notion of observations. Instead of assuming hidden causes for these observations, as in the popular partially-observable MDPs [24], we simply assume that agents can accumulate large observation and action histories. As a result, states in our MDPs are not hidden, but the state space is huge and includes (short-term) memories. We then use the size of the state space as a proxy for efficiency, and we show how to eliminate redundancy and compress the state space, while preserving the behavioural goal of the agent. Some of the mathematical theory underlying the compression of dynamical systems has been developed before in other context [25, 26], but its application to behavioral tasks and neural data is new.

We obtain two key results. First, we illustrate that model-based agents, which may seek to adjust their policy flexibly depending on context, require a different compression strategy from habitual agents, which are already set on a given policy. Second, we generate efficient representations for two standard behavioral paradigms [8, 9], and we show that the transition from sequential to persistent activity depends on the temporal basis needed to represent the task, as well as the behavioral goal (model-based versus habitual) of the agent.

## 2 From task structure to representation

A task is defined by a set of observations, a set of required actions, and their respective timing. Each trial of a task is a specific sequence (or trajectory) through the observation-action space. Any task representation is a function of these sequences, and the specific function may be defined by a RNN, or by a normative principle as in this study, that may then be compared with the trials' corresponding neural trajectories. Throughout this study, we follow the reinforcement learning (RL) framework and assume that the agent's control problem is to maximize future rewards.

### 2.1 Control under partial observability: Observation Markov decision processes

RL theory was extensively developed on the basis of Markov decision processes (MDP, [27]). In MDPs agents move through states, $s \in S$, and perform actions, $a \in A$. Given such a state and

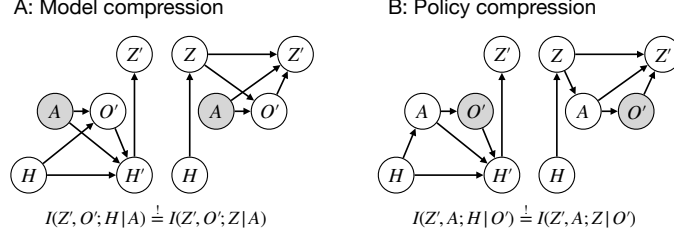

Figure 2: A: Dependency graphs for a history-OMDP (left) and a model-compressed OMDP (right). Conditioned variables are shaded. B: Same as A, but for policy compression.

action, the probability of reaching the next state, $s' \in S$, and collecting the reward, $r' \in R$, is specified by the environmental dynamics, $P(s', r'|s, a)$. An MDP is therefore defined by the tuple $\langle S, A, R, P(s', r'|s, a) \rangle$. Usually, a discount factor $\gamma$ is included, but since we are dealing with episodic problems only, we set $\gamma = 1$ for the remainder of this article. The MDP state is fully observable, meaning that the observations made by the agent at each time point fully specify the state.

More realistic tasks are partially observable, so that the agent cannot access all task-relevant information through its current sensory inputs, see Fig. 1A. A popular extension for such RL problems are the partially observable MDPs (POMDPs, [24]), which distinguish between the underlying environmental states, $e \in E$, and the agent's observations thereof, $o \in O$. Here, agents move through environmental states with probabilities $P(e'|e, a)$. In turn, they make observations $o' \in O$ (which include rewards) with probability $P(o'|e', a)$. A POMDP is therefore fully specified by the tuple $\langle E, A, O, P(e'|e, a), P(o'|e', a) \rangle$.

At each time point $t$, the environmental state $e_t$ is hidden to the agent. Consequently, the agent needs to infer this state using its action-observation history $h_t = (o_t, a_{t-1}, o_{t-1}, \dots, o_1, a_0)$, see Fig. 1B. This inference process can be summarized in the agent's belief state, $s_t \in S$, where $S = \{x \in \mathbb{R}^{|E|}_{\geq 0} | \sum_i x_i = 1\}$ is an $|E|$-dimensional simplex. The elements of this belief state are given by $s_t(e) = P(e|h_t)$. Upon taking an action $a_t$ and making an observation $o_{t+1}$, the agent can update its belief state through Bayesian inference:

$$s_{t+1}(e') = P(e'|o_{t+1}, a_t, h_t) = \frac{P(o_{t+1}|e', a_t) \sum_{e \in E} P(e'|e, a_t) s_t(e)}{P(o_{t+1}|h_t, a_t)} \tag{1}$$

Here the denominator, $P(o_{t+1}|h_t, a_t) = \sum_{e'} P(o_{t+1}|e', a_t) \sum_e P(e'|e, a_t) s_t(e)$, is the observation-generating distribution given a belief state. Formally, the state update function can be summarized by the distribution $P(s'|s, a, o')$ that equals one if Eq. (1) returns $s'$ given $s, a, o'$, and zero otherwise. On the level of beliefs, we therefore recover a MDP, called the belief MDP, defined by the tuple $\langle S, A, R, P(s', r'|s, a) \rangle$ with $P(s', r'|s, a) = \sum_{o' \in O \setminus R} P(s'|s, a, o') P(o'|s, a)$ and $O \setminus R$ denoting the set of observations excluding rewards [24].

Belief MDPs are generally hard to work with, since the belief states live on a (generally high-dimensional) simplex. Since the belief states are simply functions of the action-observation history, $h_t$, however, we could also simply use the histories themselves as states, $s_t = h_t$. To generalize this idea, we therefore define an alternative MDP directly on the level of $P(s'|s, a, o')$ and $P(o'|s, a)$ and call it observation MDP (OMDP, Fig. 1 C, given by the tuple $\langle S, A, O, P(s'|s, a, o'), P(o'|s, a) \rangle$). Importantly, $S$ may simply be chosen a discrete set.

If we choose histories as states, then the transition function $P(h'|h, a, o')$ becomes simply the append function, i.e., $h' = (h, o, a)$ (Fig. 1B,C), and only the observation function $P(o'|h, a)$ has to be specified. We will call this specific OMDP a history-OMDP for the remainder. Obviously, the history-OMDP will not be the most compact choice in general, since the set of histories grows exponentially with time, but it contains all task-relevant information, and therefore allows us to ask the question of how to compress the history space to get rid of the task-irrelevant bits.

## 2.2  State space compression

Our central goal is to find the most compact state space, $Z$, for a given task. For simplicity, we assume that the task's history-OMDP with state space $H$ is already given. As states are given by

action-observation histories, $h \in H$, we first attempt to directly find the compression function $P(z|h)$ that maps histories to compressed states, $z \in Z$, such that $|Z| < |H|$. We also define a decompression function, $P(h|z)$, by inverting $P(z|h)$ using an uninformative prior on $h$.

The compression map will depend on the specification of the type of information in $H$ that needs to be preserved in $Z$. In the following, we will distinguish two types of compression, *model compression*, which finds an efficient representation for model-based agents, which have not converged on a fixed policy, and *policy compression*, which finds an efficient representation for habitual agents, which have already inferred the optimal policy.

### 2.2.1 State space compression for model-based agents

The model-based agent needs a compression that preserves all information in $H$ about future observations [28]. In principle, the information will include observations (and their history) that may be irrelevant for a given task, but could become relevant in the future. (Once an agent has attained certainty about what is relevant or irrelevant for a given task, it should choose the more powerful compression for habitual agents, see next section). The history-OMDP's information about future observations is contained in both the observation function, $P(o'|h, a)$, and the transition function, $P(h'|h, a, o')$, and the compressed agent, with functions $P(o'|z, a)$ and $P(z'|z, a, o')$, needs to preserve information about both (Fig. 2A). Similar compressions of world-models have been studied before, see e.g. [25, 29], and we here build on these results.

Let us first consider preserving observation information when we compress the state space representation with a map $P(z|h)$. To do so, we simply require that, given any action $a \in A$, the mutual information between observations, $O'$, and either the full or compressed state space representation, $H$ or $Z$, remains the same, so that $I(O'; H|A) = I(O'; Z|A)$. Accordingly, whether we compute the next observation probability through $P(o'|h, a)$, or whether we first compress into $z$, and then compute the observation probability from there, using $P(o'|z, a) = \sum_h P(o'|h, a)P(h|z)$, should be the same.

Next we need to ensure that the compression also preserves our knowledge about state transitions. Assume we start in $h$, predict $o'$ as described above, transition to $h'$, and then compress $h'$ into $z'$. Ideally, we would obtain the same result if we start in $z$, decompress into $h$, transition to $h'$, and then compress back into $z'$. In terms of information, we thus obtain the condition $I(Z', O'; H|A) = I(Z', O'; Z|A)$. Given this constraint, we find the maximally compressive map $P(z|h)$ by minimizing the information $I(Z; H)$ between $Z$ and $H$ using the information bottleneck method [30, 31]:

$$\min_{P(z|h)} \quad I(Z; H) \quad \text{subject to} \quad I(Z', O'; H|A) = I(Z', O'; Z|A) \tag{2}$$

### 2.2.2 State space compression for habitual agents

For the habitual agent, we assume that an optimal policy, $P(a|h)$, has been obtained, and we aim to find the most compact representation of this policy. The agent thus no longer needs to predict observations, but actions. A compressed representation for a habitual agent therefore requires the transition function $P(z'|z, a, o')$ and the policy $P(a|z)$. Following the logic of the model-based agent above, we therefore need to preserve transition and action information (Fig. 2B), yielding the condition $I(Z', A; H|O') = I(Z', A; Z|O')$.

In practice, this condition requires the mutual information conditioned on observations, yet many state-observation combinations are never provided by the environment or the experimenter. An alternative and equivalent approach, which we follow here, is to preserve one-step information about actions and transitions by preserving future action sequences given future observation sequences. A trial $k$ of length $T$ is defined by the observation sequence $\{o\}_k = (o_1^k, o_2^k, \ldots, o_T^k)$ and the corresponding optimal action sequence $\{a\}_k = (a_0^k, a_1^k, \ldots, a_T^k)$. Given the history-OMDP and the policy $P(a|h)$ we can compute the likelihood of an action sequence given an observation sequence:

$$P_H(\{a\}_k|\{o\}_k) = \sum_{\{h\}} P(h_0)P(a_0^k|h_0) \prod_{i=0}^{T-1} P(h_{i+1}|h_i, a_i^k, o_{i+1}^k)P(a_{i+1}^k|h_{i+1}) \tag{3}$$

We now try to find the smallest state space representation, $Z$, with transition probabilities $P(z'|z, a, o')$ and policy $P(a|z)$, such that the action sequence likelihoods are preserved:

$$P_H(\{a\}_k|\{o\}_k) = P_Z(\{a\}_k|\{o\}_k) \quad \forall k \tag{4}$$

Here $P_Z(\{a\}_k|\{o\}_k)$ is the action sequence likelihood given the compressed representation, computed analogously to $P_H(\{a\}_k|\{o\}_k)$ in Eq. 3. Importantly, $k$ only runs over observed trials, thereby ignoring observation sequences that never occur. We use a non-parametric setting and optimize the model parameters using expectation maximization. As many state-observation combinations and thus entries of $P(z'|z, a, o')$ are encountered in none of the trials and to prevent overfitting, we put a Dirichlet prior on transitions preferring self-recurrence (see e.g. [32]). Furthermore, we find the smallest state space $Z$ by brute-force. Specifically, we initialize the model with different $|Z|$, optimize the model parameters, and then take the smallest model that fulfils the likelihood condition 4.

### 2.3 Towards a more biologically realistic setting: Linear Gaussian OMDP parametrization

So far we have discussed the discrete or non-parametric treatment of tasks using discrete OMDPs. As we will show below, the non-parametric case can already give us several conceptual insights on task representations. However, to become more realistic and deal with real-valued neural activities, continuous observation spaces, and the noisiness of the brain, we need to look at possible parametrizations. Here we discuss a linear parameterization that allows us to intuitively interpret the model and make several connections to neural properties and network dynamical regimes. Furthermore, by introducing representation noise we can describe trade-offs between accuracy and complexity of representations, given a limited capacity. This automatically compresses the state space for efficiency reasons, as we will show below.

We only consider the habitual agent here, for brevity, but a model-based agent with a full OMDP model can be modelled analogously. In the non-parametric case the model parameters were parameters of categorical distributions. Assuming an $N_z$-dimensional state vector $z \in \mathbb{R}^{N_z}$, we here parametrize the model with normal distributions:

$$\begin{aligned}
P(z'|z, a, o') &= \mathcal{N}(Az + B_a a + B_o o', \sigma_t^2 I) \\
P(a|z) &= \mathcal{N}(Cz, \sigma_r^2 I).
\end{aligned} \tag{5}$$

Here, $A \in \mathbb{R}^{N_z \times N_z}$ is the transition matrix, $B_a \in \mathbb{R}^{N_z \times N_a}$ and $B_o \in \mathbb{R}^{N_z \times N_o}$ are the input weights of past actions and observations, respectively, $C \in \mathbb{R}^{N_a \times N_z}$ are the weights of the readout (here the policy), and $\sigma_t$ and $\sigma_r$ are scalar standard deviations of the isotropic transition and readout noise, respectively. Our system therefore corresponds to a linear dynamical system (LDS) for the state $z$. We will set the readout noise to zero for the remainder as we are only interested in how transition noise accumulates over time, modelling memory decay over time. Since there is a degeneracy in the scaling of the parameters $A, B_a, B_o, C$, and the transition noise, $\sigma_t$ (see e.g. [33]) which allows the system to get rid of noise trivially, we constrain the state values from above and below so that $0 \leq \mu(z(i)) \leq z_{\max}$ for all $i = 1 \dots N_z$.

Given this limited capacity, both task-relevant and task-irrelevant information have to compete for resources. Accordingly, policy-irrelevant information will be ignored in favor of an accurate representation of relevant information, thus leading to compressed representations. We discuss this intuition in more detail in the Supplementary Material, and we exemplify in the simulations, below. Finally, we optimize the LDS by maximizing the likelihood of the target policy with respect to parameters $A, B_a, B_o, C$, analogous to the non-parametric policy compression case before.

## 3 Compressed state space representations and neural activities

### 3.1 Non-parametric policy compression for a delayed licking task

We will first apply our non-parametric policy compression on a delayed directional licking task in mice [34, 9]. In this task, mice have to decide whether a tone is of low or high frequency, and then report their decision, after a delay, by licking one of two water delivery ports. We model two versions of this task, one with a fixed delay period (fixed delay task, FDT, Fig. 3A-E) and one with a randomized delay period (random delay task, RDT, Fig. 3F-H). Neurons recorded in the ALM (anterior lateral motor cortex) show a striking distinction between the tasks: while activity changes during the delay period in the FDT, it remains at a steady level in the RDT [9].

A key difference between the two tasks is that the timing of the go cue is unpredictable in the RDT, but predictable in the FDT. A predictable go cue allows the animal to prepare its action, which we will model by introducing a sequence of preparatory actions (e.g. open mouth, stick tongue out, or

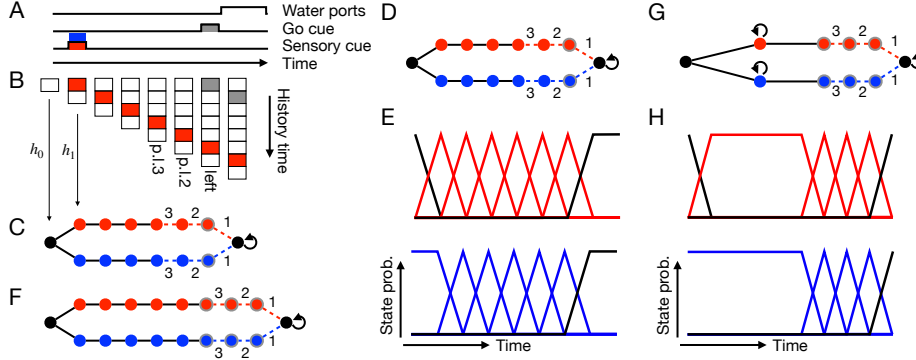

Figure 3: A-E: Fixed delay task (FDT). F-H: Randomized delay task (RDT). A: Task structure. Each trial starts with a tone (red or blue) that indicates the reward location. Rewards are available after a go cue (grey) that arrives either after a fixed (FDT) or randomized (RDT) delay. B: Example trial (red tone) and sequence of corresponding history states (columns, only sensory and go cues are shown). Underneath each history state, the corresponding optimal action is indicated (blank means action wait, p.l stand for preparatory left, and so on). C: History task graph with optimal policy. Nodes are history states (post-go cue states have a grey rim), edges are actions. Dashed red edges correspond to (preparatory) left actions, and dashed blue edges to (preparatory) right actions. D: Task graph for FDT after compression of the optimal policy. E: State probabilities for the two trial types. F: Same as in C, but for the RDT. G: Task graph for RDT after compression. H: as in E, for a given delay length.

internal preparations) before the actual left or right licking action (Fig. 3B). Furthermore, we assume that the agent takes decisions as fast as possible, in order to maximize its reward consumption. In turn, the resulting optimal policy for the FDT initiates the action sequence before the go cue (Fig. 3C) while in the RDT the sequence is initiated after the go cue (Fig. 3F). These differences are reflected in the resulting compressed state space representations shown in Fig. 3D and G, respectively.

In the FDT, the task representation keeps precise track of time during the delay period (Fig. 3D). Each time point effectively becomes its own state, and the model sequences through them. If we identify each state with the activation of an individual neuron (or, more realistically, of a population mode), then neural activities turn on and off as in a delay line (Fig. 3E). This task representation thereby allows the agent to take the preparatory actions before the onset of the go-cue. We note that recorded neural activities are generally slower (they 'ramp' up or down) than the fast delay line proposed here. Such 'ramping' provides a less precise (and thereby 'cheaper') encoding of time which may be sufficient for this task as the gain for precise timing is only minor (faster access to reward). Here we only consider compressed representations that preserve future returns, and do not consider possible tradeoffs between the future returns and the compressed representations. These idealized representations require a fast delay line.

In contrast, the compressed state representation of the RDT combines all delay states and thereby discards timing information (Fig. 3G). In turn, the (compressed) state does not change during the delay (Fig. 3H). This representation is sufficient to represent the optimal RDT policy.

## 3.2 Non-parametric and linear compression for a somatosensory working memory task

Next, we study model and policy compression in a (somatosensory) working memory task in monkeys [8], see Fig. 4A. In this task, each trial consists of two vibratory stimuli with frequencies $f_1$ and $f_2$ that are presented to a monkey's fingertip with a 3sec delay. To get a reward, the monkey has to indicate which of the two frequencies was higher. Neural activities in the prefrontal cortex recorded during the task show characteristic, temporally varying persistent activity during the delay period, as observed for many other working memory tasks [35], see also Fig. 5A,C.

The history-OMDP of this task is shown in Fig. 4B. When compressing the history space using the method for the model-based agents, we find that all states during the delay period remain uncompressed, as they are predictive of the $f_2$ observation. After $f_2$ is observed, history states with the same action-reward contingencies are combined in the compressed representation, yielding only

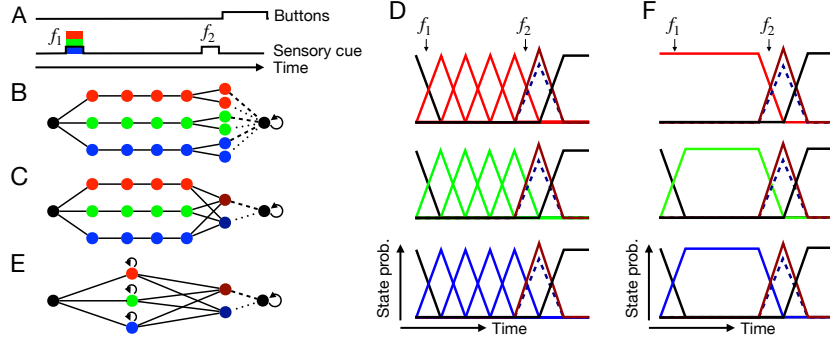

Figure 4: A: Task structure. We only model three $f_1$ frequencies for simplicity, coded red, green and blue. B: Task graph based on history states for the optimal policy (constructed as in Fig. 3). C: Task graph for model compression. States requiring the left and right actions are combined into a single dark red and dark blue state, respectively. D: State probabilities over time after model compression for all six trial types. Rows correspond to different $f_1$ values. E: Task graph for policy compression. Delay states after the $f_1$ presentation are combined. F same as D, but for policy compression.

two states ($f_1 > f_2$ and $f_1 < f_2$), which effectively correspond to the subject's decision (Fig. 4C). If we again identify each state with the activation of a neural population mode, we find a component corresponding to the decision, as observed in the data [35], but also a precise encoding of time during the delay period which does not reflect recorded activity (Fig. 4D).

Animals well-trained on tasks may be assumed to behave habitually. Indeed, when we seek to only preserve policy information, and when we assume that the animal is not preparing any actions during the delay period, we find that we can compress the state space even further (Fig. 4E,F). All delay states corresponding to different $f_1$ frequencies are merged, so that any timing information is lost. When looking at the state representation over time, we find persistent activity (Fig. 4F), just as in the RDT above (Fig. 3G). The persistent state dynamics here contrast with the sequential state dynamics of the FDT above (Fig. 3E). While in both tasks the delay is fixed, in the directional licking task a decision is stored while here a stimulus is stored and (under the assumption that no action needs to be prepared) timing during the delay is irrelevant.

While the non-parametric treatment yields several conceptual insights, it does not allow for a direct comparison with data. For instance, the delay line activity of the model-based agents crucially depends on the time step of the simulation, and assumes a completely noise-free evolution of the internal representations. To move closer to realistic agents, we finally model the somatosensory working memory task using the parametric LDS approach, which also includes noise. A trial is structured as in Fig. 4A, but with $\{f_1, f_2\} \in \mathbb{R}$ being continuous scalars. Given the rigidity of the linear parametrization we make a couple of simplifying assumptions: First, we only maximize the accuracy in the actual decision (left or right) and ignore previous actions altogether. The transition function then also becomes action independent, i.e., we set $B_a = 0$. Second, we approximate the (nonlinear) decision function, $d = \text{sign}(f_1 - f_2)$, with a linear function, $y = f_1 - f_2$.

The accuracy of the representation is thus fully defined by the readout distribution, $P(y|h_{T_D}) = N(\mu_y, \sigma_y^2)$, at decision time $T_D$, right after $f_2$ is observed. The mean, $\mu_y = c^\top \mu(z_{T_D})$, and variance, $\sigma_y^2$, of this readout are functions of the mean and variance of the final state $z_{T_D}$, which can be computed by unrolling the LDS. Specifically, $c^\top \in \mathbb{R}^{N_z}$ is the readout vector, and the final state mean, $\mu(z_{T_D})$, is computed by $\mu(z_{T_D}) = \Phi h_{T_D}$, with $\Phi = \begin{bmatrix} B_o & AB_o & \dots & A^{T_D-1}B_o \end{bmatrix} \in \mathbb{R}^{N_z \times T_D}$ being the linear map from histories to compressed states, analogous to $P(z|h)$ in the non-parametric case above. The compressed state space can thus be understood as a linear subspace in the space of all histories, defined by $\Phi$. We finally find this subspace by maximizing the likelihood of $y = f_1 - f_2$ given $h_{T_D}$ with respect to $A, B_o, c$ as described in section 2.3. Simulation details, parameter values and code are provided in the Supplementary Material.

The resulting state representation dynamics resemble brain activity well on a single neuron level (Fig. 5A,B) as well as on a population level (Fig. 5C,D). Furthermore, the state dynamics are low-dimensional, a sign of the successful compression (Fig. 5D). Indeed, when looking at the linear map

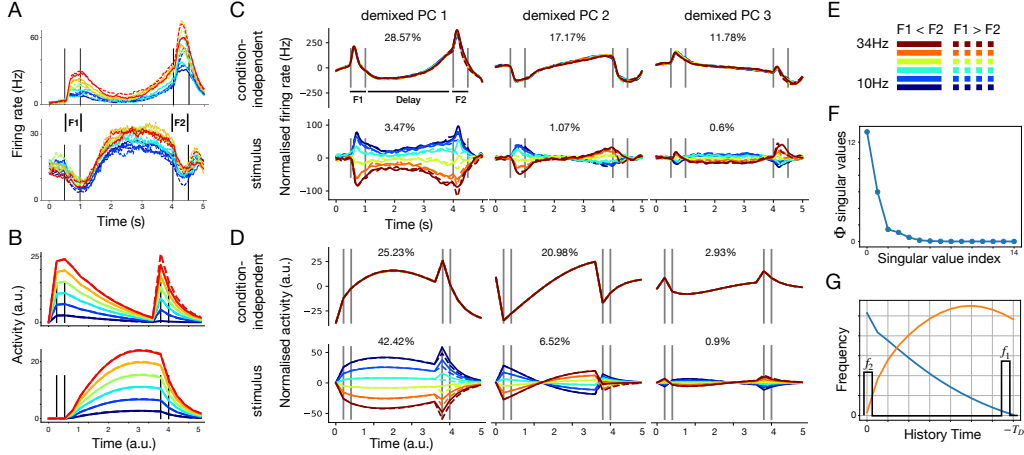

Figure 5: A: Peristimulus time histograms of two PFC neurons. Lines follow legend shown in E. B: Two matching model neurons (i.e., two state dimensions of $z$). C,D: Population level comparison using demixed principal component analysis [35]. We demixed condition-independent variance and stimulus dependent variance. C: First three condition-independent components (first row) and stimulus components (second row) of PFC neurons. D: The corresponding components of the model. Fraction of explained variance is indicated on top of each component. As components may be non-orthogonal, they do not have to add up to 100%. F: Singular values of $\Phi$. G: The history space with one example history, $h_{T_D}$, drawn in black, underlaid by two vectors in $\Phi$'s row space. Specifically, since the readout weights $c$ compute the $f_1 - f_2$ difference from the state $\mu(z_{T_D}) = \Phi h_{T_D}$, and since $z_{T_D} \geq 0$, some entries of $c$ are positive ($c_+$), some negative ($c_-$). The blue and orange lines correspond to $-c_-^\top \Phi$ and $c_+^\top \Phi$, respectively. All neural data was processed as described in [35].

$\Phi$ directly, we see it dominated by two dimensions (Fig. 5F). Specifically, these two dimensions divide the history space in two bins, one for recent observations, i.e. $f_2$, and one for observations in the past, i.e. $f_1$ (Fig. 5F). Timing information of the stimuli is thus compressed away, similar to the non-parametric case (Fig. 4E,F).

## 4 Discussion

In this article we have proposed a new, normative framework for modeling and understanding higher-order brain activity. Based on the principle that neural activity reflects a maximally compact representation of the task at hand, we have reproduced dynamical features of higher-order brain areas for two example tasks involving short-term memory, and we have explained how those features follow from the normative principle.

The key principle underlying this work— representational efficiency—has been proposed before in various context. For instance, the efficient coding hypothesis has held that redundant information in sensory inputs should be eliminated [23]. Information-theoretical considerations have led to the proposal that the brain should only keep information about past events that is relevant for maximizing future returns [36], which naturally suggests some combination of efficient coding and reinforcement learning [37]. Moreover, indirect evidence for efficient task-representations has been found in the activity of dopaminergic neurons [38]. We were also inspired by considerations of efficient representation, or coarse-graining, of dynamical systems [26].

On a technical level, our work extends previous studies that have considered RL under costs. While we have focused on representational costs, previous work has studied RL under control costs [39, 40]. Our approach also extends previous work on representation learning for RL [25, 29, 41, 42, 43]. While [42, 43] consider simultaneous learning of representation and control, we have not considered the problem of learning. In theory, an agent could first learn a history-OMDP model, from there solve, or plan, for the optimal policy using dynamic programming, and then compress the policy. This learning strategy of going from a model-based strategy to a model-free strategy has been conceived before [44, 45, 46]. In practice, starting with the full and detailed history representation will often

prove infeasible, and one would therefore assume that agents also have to go the opposite way: starting with a coarse representation that is then expanded [46]. Consequently, there are many paths conceivable on how to get to a compact representation, each of which might have different advantages. We therefore consider learning a separate, and presumably more difficult problem, and leave it for future work.

Finally, we think that the OMDP framework and especially parametrizations thereof might be a fruitful avenue for partially observable RL research. POMDPs have been shown computationally untractable and new ways of dealing with partial observability are considered to be needed (see e.g. [27], chapter 17.3). Furthermore, RNN systems used for RL, such as in [47], are effectively using OMDP parametrization.

## Broader Impact

The study presented here is aimed at resolving some of the current debates in the field of working memory and decision making. In that sense, our work has the potential of impacting and progressing this field mainly conceptually. We note that our work does not seek to push the state of the art of machine learning in terms of performance. The non-parametric methods used here are limited and do not scale up to larger architectures, and their benefits lie in clear interpretability rather than performance. We believe that a better understanding of decision-making circuits, whether biological or artificial, may eventually benefit the safety of computational learning architectures.

## Acknowledgments and Disclosure of Funding

This work was supported by the Simons Collaboration on the Global Brain (543009) and the Fundação para a Ciência e a Tecnologia (FCT; 032077). SB acknowledges an FCT scholarship (PD/BD/114279/2016). The authors declare no competing financial interests.

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
