[Supplementary Material]

# Compact task representations as a normative model for higher-order brain activity: Supplementary material

**Severin Berger**
Champalimaud Centre for the Unknown
Lisbon, Portugal
severin.berger@neuro.fchampalimaud.org

**Christian K. Machens**
Champalimaud Centre for the Unknown
Lisbon, Portugal
christian.machens@neuro.fchampalimaud.org

## 1  History representation under a noisy, constrained linear dynamical system

Here we give a short analysis of how different linear dynamical systems (LDS) result in different tradeoffs of representing relevant versus irrelevant parts of a history, given transition noise. We will use the same notation as in the main text.

Let us look at a single scalar readout $y$ of an arbitrary LDS, which, at time t, is given by:

$$y_t \sim N(c^\top \mu(z_t), c^\top \sigma^2(z_t) c) \tag{1}$$

Here, $c$ correspond to the readout weights. The mean, $\mu(z_t)$, and the covariance, $\sigma^2(z_t)$, of the state variable, $z_t$, are given by (under the assumption of certain initial state conditions $\mu(z_0) = 0$ and $\sigma^2(z_0) = 0$):

$$\mu(z_t) = \Phi h_t$$
$$\sigma^2(z_t) = \sigma_t^2 \sum_{n=0}^{t-1} A^n (A^n)^\top \tag{2}$$

with

$$\Phi = \begin{bmatrix} B_o & AB_o & \dots & A^{t-1}B_o \end{bmatrix} \tag{3}$$

The dynamics and input weights are denoted by $A$ and $B_o$, respectively. As noted in the main text, there is a degeneracy between the scaling of the transition noise $\sigma_t$ and the weights of the LDS. We chose to constrain this degeneracy by bounding state values from below and above. Here we want to first describe our rationale behind this choice of constraint. Second, we describe how an LDS under this constraint must trade-off between relevant and irrelevant information.

### 1.1  Constraining the LDS to remove scaling degeneracy

As seen in the equations above, the uncertainty in the readout is determined by two parameters, the readout vector $c$ and the dynamics $A$. The input weights, $B_o$, only feature in the mean, it is thus easy to see that the readout noise can be eradicated by decreasing $\|c\|$ arbitrarily and increasing $\|B_o\|$ accordingly, while leaving the readout mean unchanged. This leads to a first degeneracy, independent of the dynamics $A$, that quenches the noise trivially. A second degeneracy may arise through transient

amplifications by non-normal dynamics. An example would be a purely feedforward dynamics that amplifies the input.[1]

Common to both degeneracies is that they amplify the input such that the additive transition noise, $\sigma_t$, is relatively scaled down. Hence, both degeneracies may be removed by constraining the scale, or some norm, of the state $z$. We chose to bound each dimension separately for the following reasons:

Each state dimension $z_i$ can be interpreted as a component in the representation of a history. Specifically, $\mu(z_t) = \Phi h_t$ and thus an estimate $\hat{h}_t$ of the history can be obtained through $\hat{h}_t = \Phi^\dagger \mu(z_t)$, where $\Phi^\dagger$ denotes the Moore-Penrose inverse of $\Phi$. Each column $\Phi_i^\dagger$ thus defines the direction in history space that component $z_i$ is representing. Since we have $N_z$ such components we can represent $N_z$, potentially overlapping, directions in the history space. By bounding the activity of each component, or state dimension, separately, as opposed to bounding the $L_1$ or the $L_2$ norms of the state vector, we effectively assume all components to be independent, i.e. if one component is more active it does not imply that another component has to be less active. For example, if only one history direction is task-relevant, then all $N_z$ components can code for the same direction, reducing the uncertainty in the representation of that direction because the readout may average $N_z$ representations. If there are $N_z$ independent relevant directions in the history space, every component has to code for a separate direction, thus effectively increasing the noise. This trade-off, that follows from the limited capacity of the constrained LDS, is discussed in more detail in the following section.

## 1.2 Representational tradeoffs under limited capacity

As described above, our LDS has a limited capacity defined by the number of components $N_z$ and the range of activity $[0, z_{\max}]$. Furthermore, both degeneracies will scale up the system state to effectively scale down the relative size of the transition noise $\sigma_t$, thereby pushing the system's dynamic range to have $z_{\max}$ as an upper bound. Here, we first describe how such a system has to focus on relevant information only in order to increase the precision in the representation of this relevant information (Fig. 1A). Second, we illustrate how a system may also increase precision by ignoring timing information, if irrelevant (Fig. 1B).

Let us consider a single state dimension $\mu(z_i) = \phi_i^\top h$, where $\phi_i^\top$ is the $i$'th row of $\Phi$ and $h$ is a history. Due to the upper bound $z_{\max}$ we have:

$$
\begin{aligned}
z_{\max} \geq \mu(z_i) &= \phi_i^\top h \\
&= \cos\alpha \|\phi_i\| \|h\|
\end{aligned}
\tag{4}
$$

As the length of $\phi_i$ will scale any signal in $h$, $\|\phi_i\|$ effectively determines the signal to noise ratio. More specifically, each component can either represent the whole history faithfully ($\alpha = 0$) with low precision $\|\phi_i\| = z_{\max}/\|h\|$, or only parts of the history $\alpha > 0$ with higher precision $\|\phi_i\| > z_{\max}/\|h\|$. (For simplicity we assumed here that $h$ is the history hitting $z_{\max}$, i.e. $z_{\max} = \phi_i^\top h$). Figure 1A illustrates this trade-off for a single represented history.

The same trade-off applies to the representation of two histories adjacent in time that both contain relevant information. Specifically, we consider $h_1 = \begin{bmatrix} h & 0 \end{bmatrix}^\top$ and $h_2 = \begin{bmatrix} 0 & h \end{bmatrix}^\top$, and assume that only the magnitude $h$ is task-relevant, but not at what time $h$ was observed. Representations ignoring timing information then increase precision (Fig. 1B). Ignoring timing information generally leads to a smoothing of the representation over time.

As a note of caution we emphasize here that the illustrations in Figure 1 and the explanations given here aim at describing the main effects governing the representation of histories, but they do not represent a rigorous analysis. For example, even though we treated rows $\phi_i$ as independent, the matrix $\Phi$ is not any arbitrary matrix, but a matrix following the specific structure in equation 3. Furthermore, this structure depends on the dynamics $A$ which also influences the noise accumulation, and, as noted above, we ignore this interaction. Nevertheless, we found the intuitions presented here quite useful and accurate, as for example seen in Figure 5 of the main text.

Figure 1: **Trade-offs in a noisy, constrained LDS.** A: We consider a history $h = h_{\text{rel}}e_{\text{rel}} + h_{\text{irr}}e_{\text{irr}}$ consisting of a relevant direction $e_{\text{rel}}$ and an irrelevant direction $e_{\text{irr}}$, scaled by $h_{\text{rel}}$ and $h_{\text{irr}}$, respectively. All three vectors are assumed to have unit length. We consider two choices of represented directions, $\phi$ (blue) and $\phi^*$ (green). Their respective angles to the history are $\alpha$ and $\alpha^*$. The lengths $\|\phi\|$ and $\|\phi^*\|$ are constrained by their projection onto the history $h$, specifically by the upper bound $z_{\text{max}}$ (dashed titled line). As a longer $\phi$ in general signifies lower noise, dashed half circles are depicted akin to noise isoclines. Direction $\phi^*$ leads to a longer projection onto the relevant direction (red-green dashed arrow) than $\phi$ (red-blue dashed arrow), thus leading to a higher precision representation of the relevant information. The increased precision is thus enabled by discarding irrelevant information. B: Representation of two histories adjacent in time, specifically $h_1 = \begin{bmatrix} h & 0 \end{bmatrix}^\top$ and $h_2 = \begin{bmatrix} 0 & h \end{bmatrix}^\top$. Both histories have unit length. We assume only the magnitude of $h$ is relevant, but not when in the history $h$ was observed. We consider two sets of two representation directions, $\phi_1, \phi_2$ and $\phi_1^*, \phi_2^*$. Angles and constraints are depicted as in A. The first set, $\phi_1, \phi_2$, codes for both directions almost orthogonally (small $\alpha_1$, small $\alpha_2$). Each history is thus represented faithfully. In the second set, $\phi_1^*$ and $\phi_2^*$ have reduced overlap with $h_1$ and $h_2$, respectively, and thus no longer represent the two histories faithfully. Specifically, $\phi_1^*$ and $\phi_2^*$ give up timing information, but both still represent the task-relevant magnitude $h$. In fact, they represent $h$ at both time steps, the readout can therefore average the two representations to reduce readout noise.

## 2 Description of the LDS model of the somatosensory working memory task

Here we describe in more detail the LDS model of the somatosensory working memory task [1] introduced in section 3.2 and Figure 5 of the main text.

### 2.1 Model setup and parameter choices

In our model, a trial consists of $T_D = 15$ time steps. The first frequency, $f_1$, is presented on the second time step, the second frequency, $f_2$, on the last time step. Thus all entries of the history $h_{T_D}$ at decision time $T_D$ will be zero, except for $h_{T_D,1} = f_2$ and $h_{T_D,14} = f_1$. Since $h_{T_D} \in \mathbb{R}^{15}$, we set $N_z = 15$ in order to give the system the capacity to represent the history space faithfully. The upper bound $z_{\text{max}}$ will be reached by the history $h_{T_D}^{\text{max}}$ of maximum length. Assuming $\|h_{T_D}^{\text{max}}\| = 1$, we set $z_{\text{max}} = 1$. Lastly, we set the transition noise variance $\sigma_t^2 = 0.01$.

### 2.2 Optimization

As described in the main text, we maximize the log-likelihood of the $f_1 - f_2$ difference $y$, given the history $h_{T_D}$, subject to the state values being within the lower (0) and upper ($z_{\text{max}}$) bounds. The

likelihood, for a trial $i$ with frequencies $f_1^i$ and $f_2^i$, is given by:

$$P(y_i|h_{T_D}^i) = N(\mu_{y^i}, \sigma_{y^i}^2) \tag{5}$$

Using equations 1 and 2, we have $\mu_{y^i} = c^\top \mu(z_{T_D}^i) = c^\top \Phi h_{T_D}^i$, and $\sigma_{y^i}^2 = c^\top \sigma^2(z_{T_D}^i)c = \sigma_t^2 c^\top \sum_{n=0}^{t-1} A^n (A^n)^\top c$. We thus see that the readout variance is trial independent and we write $\sigma_y^2$. The log-likelihood $L(y_i|h_{T_D}^i)$ is then given by:

$$L(y_i|h_{T_D}^i) = -k - \log(\sigma_y) - \frac{(y_i - c^\top \Phi h_{T_D}^i)^2}{2\sigma_y^2} \tag{6}$$

Here $k$ is a constant. To speed up optimization, we approximate the trial-dependent term $(y_i - c^\top \Phi h_{T_D}^i)^2$ by first writing $f_i = \begin{bmatrix} f_1^i & f_2^i \end{bmatrix}^\top$, and then $y_i = \begin{bmatrix} 1 & -1 \end{bmatrix} f_i$ and $h_{T_D}^i = M f_i$, where $M \in \mathbb{R}^{T_D \times 2}$ is the appropriate matrix mapping $f_1^i$ and $f_2^i$ to the history $h_{T_D}^i$. We then have $(y_i - c^\top \Phi h_{T_D}^i)^2 = (\delta^\top f_i)^2$, with $\delta^\top = \begin{bmatrix} 1 & -1 \end{bmatrix} - c^\top \Phi M$. The log-likelihood is thus maximal for every trial if $\|\delta\|^2 = 0$, i.e. when the composition of the readout $c$ with the map $\Phi$ performs the $f_1 - f_2$ operation. We thus approximate $(\delta^\top f_i)^2$ by $\|\delta\|^2$, thereby making the log-likelihood trial independent. Finally, we optimize using gradient ascent.

### 2.3 Generating model sequences to compare to neural data

After optimizing the LDS as described above, we want to compare the resulting state trajectories to the corresponding neural trajectories.

To generate such state trajectories, we run our optimized LDS providing both positively and negatively scaled $f_1$ and $f_2$ frequencies, as is common among other models of this task, e.g. [2, 3], and as motivated by the frequency coding in the secondary somatosensory cortex S2 [4], the structure providing the frequency information to the PFC. Specifically, after generating a state sequence for trial $i$ with frequencies $f_1^i$ and $f_2^i$, we generate a second sequence with frequencies $f_{1-}^i$ and $f_{2-}^i$, where $f_-^i = a + (b - f^i)$. The scalar $b$ is set to the maximum of all frequencies presented, and $a > 0$. Each trial's modelled state sequence thus has $2N_z$ dimensions.

Next we modify the state trajectories after $f_2$ is presented. Neural trajectories move back to baseline after the $f_2$ presentation and the decision, thereby discarding for example frequency information. This makes sense from our efficiency perspective, but is not within the scope of our LDS model. In order to not let the principal dimensions of our model data be affected by this post-$f_2$ activity, we artificially let the state decay after $f_2$ presentation.

Both steps described here, as well as all the steps described in section 2, are implemented in the provided code.

## Footnotes

[1] We note that dynamics amplifying the input will in general also amplify the noise. Without making any structural assumptions on $A$, such as assuming normal dynamics, it is hard to treat this intricate relationship analytically. We thus content ourselves with noting that dynamics that trivially quench the noise exist, and thus must be constrained.