[Reviews · NeurIPS 2020]

Review 1

Summary and Contributions: In this paper, the authors suggest building a compressed history representation of state to solve partially observable MDPs. They look at two limits - a 'model-based' one in which it is necessary to predict from the compressed all future observations; and a 'model-free' one in which it is only necessary to have the information that will disambiguate the actions. They also consider a linear Gaussian state-space model. They apply the method to a cartoon of a licking task for a mouse and to a flutter disrimination task in monkeys.

Strengths: The paper nicely melds some ancient (utile distinctions); middle-aged (information bottleneck) and more modern themes. It is good to see the combination of informational and RL ideas; and it was also important that the paper did not only consider the rather impracticable non-parametric methods, but also the much simpler state-space model. The applications to licking and flutter are good exemplars of the method.

Weaknesses: One main problem is that the paper does not contain a plausible method for learning. Not only would this likely be extremely hard (for the informational measures), but there could also be a complex interaction between things like compression, exploration and learning. it was notable that the information theoretic cost function of the non-parametric approach was abandoned for a maximum likelihood method for the state-space model. [Post author response: I don't agree that it necessarily makes sense to consider a representation without paying attention to learnability -- as I noted, the requirement for active sampling makes this almost recursively complex.] Although it is certainly interesting to think about the difference between model-based and model-free representations, I wasn't completely convinced by the arguments in the paper. If I understand correctly, the habitual agent would have a partly open-loop character to it (ie it would ignore parts of the observation) - this is dangerous in anything but a completely stationary world; and since animals seem to continue to possess their model-based methods even after control has become habitized, it would also seem that the suggestion would be that animals would maintain two separate representations, one MB and the other MF, which seems wasteful. The experiments could also have been more convincing. First, mice are often promiscuous lickers even in the RDT -- could the authors estimate exactly how much sooner they get the reward in the FDT (noting the noise in interval timing). Second, the extended ramping up dynamics in the flutter task towards F2 is missing in the model. [Post author response: the PFC neuron in 5A top starts ramping upwards after about half the interval; the model neuron in 5B top starts ramping up right before the end of the interval; the authors suggest comparing the demixed PCs - demixed PC1 is a nice case in point here showing these different dynamics nicely.] It would have been interesting to have explored the tradeoff that Tishby and Polani have considered - not enforcing the informational constraints strictly, but having a tradeoff of information for reward - a bit like rate distortion theory.

Correctness: yes - the paper looked fine. The problem with the habitual agent version - the lack of state-observation combinations - could have been explained better. The method that is proposed conditions on the actual observations - but, presumably, there need to be enough of these to cover the space well - some analysis/experiments with this would have been welcome.

Clarity: yes

Relation to Prior Work: Quite a bit of prior work is mentioned (although I didn't see a reference to prediction suffix trees?). However, very little of it was actually discussed - it would have been worth spending space on this.

Reproducibility: Yes

Additional Feedback:


Review 2

Summary and Contributions: This paper proposes a normative method to compress the state space under a fully-observed or partially observed MDP framework, eliminating task-irrelevant variables. The authors considered state space compression for model-based agents and agents with a fixed optimal policy. The methods are applied to two tasks, and the authors verified that the compressed states dynamics resemble brain activities on the level of single neuron as well as population.

Strengths: Good task representations are important both for making decisions, and for understanding how the brain makes decisions. Conceptually this work aims to solve derive efficient representations, and has a nice conceptual formalism to explain what needs to be compressed. In addition, the paper combines the idea of efficient coding by eliminating task-irrelevant information during state compression. The authors also evaluated the method by comparing the results with existing work for neural analysis. With the non-parametric method proposed in the paper, the compressed states have a similar dynamic pattern as that shown in the neural activities.

Weaknesses: As a conceptual model, this approach is clear and interesting, but I don't see the conceptual novelty. It is already known that belief states can be compressed through the information state about the world (Bertsekas 1995) and about the value (Poupart and Boutilier 2003). As a practical model, the authors make no claim about their method's general utility, and only apply it to a very simple task where they can use a brute-force exhaustive compression strategy. In section 2.2.2, the likelihood condition(eq.4) for the the state compression for habitual agents is confusing. This equation is marginalized probability over history h or over the compressed state z, and should always hold, regardless of the choice of z. I am not sure how this equation can serve as a criterion to find the compressed state. [Update after rebuttal: the authors understand the concern, but I have no idea what their remedy is. They talk about a different factorization, but that should not fix the concern.] The dynamics of the compressed states z do resemble that in the neural activities. But it would be more convincing if there can be a comparison of the dynamic of the compressed state z and that of the uncompressed state h to reveal what information has been preserved and/or eliminated. Is this actually a finding that results from the author's approach, or is that generic property for any belief state? I expect that invertible transformations of the compressed space could easily look different than the neurons, in which case what is it that favors a representations that resembles the biology? [Update after rebuttal: The authors point us to a few figures distinguishing pre- and post-compression, thanks for the clarification. They do not address my concern about invertible transformations changing similarity to the neural representation.]

Correctness: yes

Clarity: moderately well written

Relation to Prior Work: The novelty of any conceptual advance here should be clarified. [update after rebuttal: I think the authors understate the prior work on efficient representation for actions, ie policy compression. Perhaps they are focused on reinforcement learning, where the compression is indeed more naturally in the state space. But there's still work on policy compression in optimal control, where learning and adaptability are not the focus.]

Reproducibility: Yes

Additional Feedback: Upon further consideration, the interest in relating to neural representations is sufficient to justify increasing my score from 5 to 6.


Review 3

Summary and Contributions: This manuscript addresses a fundamental question in computational neuroscience: what computational principles give rise to neural activity in higher order brain areas and through what lens can we interpret them? It does so by following a normative approach, coupling principles of efficient coding (historically applied to the domain of vision science, but here applied in a novel capacity to account for higher order neural phenomena) with the framework of Markov Decision Processes. The major result is to illustrate through derivation how task state spaces can be compressed in distinct ways according to two different behavioural goals. That is, distinct compression will arise depending on whether the agent behaves habitually having converged on a fixed policy (and so may compress their policy directly, ​policy compression​), or whether the agent may seek diverse future policies within a fixed world model, in which case the policy cannot be usefully compressed, but the world model can be (​model compression​). Having derived a normative non-parametric solution to this problem, the authors show that with some minor but valuable adjustments these distinct compression strategies differentially predict low level neural phenomena in two tasks with different model species (mice and monkeys). The contributions are conceptually impactful, and provide a novel scientific tool with which to predict neural motifs from normative principles. The work is likely to inspire many future theoretically-driven neural experiments. Please note that I have revised my numerical score down to a 7 post-rebuttal due to an error in my reading of the scoring system.

Strengths: - While normative approaches have been commonly applied in psychology to explain behavioural phenomena, and efficient coding techniques applied to account for low level neural responses in vision neuroscience, it is much less common (but especially impactful) to see a model presented which span these levels by predicting different qualitative neural motifs from different behavioural strategies. This is a major strength of this work and places this manuscript in a unique position in the literature (along with few other approaches, such as RL/reward prediction errors which have become a particularly powerful framework for neuroscience). - Testing the predictions on two different tasks which used different model systems was also appreciated and relatively rare in computational neuroscience. - The theoretical grounding appears to be solid, novel for this application and was well explained.

Weaknesses: I have only two minor comments here: - The authors contrast their approach with RNN modeling of neural responses and state that “training a RNN does not clarify why a particular solution is a good solution, or, indeed, if it is a good solution at all”. I appreciate that the two approaches have different advantages but see them as complimentary, and I am not sure that framing RNNs in opposition to normative theories draws a useful distinction. As the authors point out, the parameterisation of their nonparametric approach corresponds to a linear dynamical system which has close connections to RNNs. - In Figure 3, its not clear to me that these neural predictions are uniquely predicted by their normative account. Would an alternative agent, similarly defined in the RL setting but without history compression predict the same neural responses illustrated in Fig 3? It seems as though Figure 3 shows the same diagram for the FDT task whether or not there is policy compression.

Correctness: The methodology is novel for the neuroscience community and both mathematically and empirically sound.

Clarity: This is an especially clear and well-written manuscript in which the authors clearly explain their approach, assumptions, predictions and results, whilst remarkably packing their considerable theoretical contributions into the required 8 pages. I would happily read an extended version of this work and highly recommend it for publication.

Relation to Prior Work: Yes, the authors clearly place their work in the context of the experimental and computational neuroscience literatures.

Reproducibility: Yes

Additional Feedback: - The authors mention the benefits of compression for resource-efficient representation in the brain, however compression has additional implications for how an agent will generalise their behaviour (and the neural underpinnings of that behaviour) to new environments. Abstracting away from the specifics of an experience could make an agent more noise-tolerant, or generalise in specific ways based on the way their knowledge of the world, or policy was compressed. Does the technique in this paper give rise to specific neural or behavioural predictions for an agent exposed to new tasks? While I appreciate the authors have little room to add more content, a comment along these lines might help to direct follow-up research in the wider community. - While the authors explicitly state that they do not consider the learning problem in this work, one can’t help but consider what the implications of compression (be it model compression or policy compression) would be on the trajectory of learning. Intuitively, if an agent were to compress too soon, information will be lost that could potentially become relevant only later through further exploration and learning. Investigating this point would be outside of the immediate scope of this work, but this is probably one of many ways this work could inspire the larger research community. - The legend of Figure 3 needs some expansion as there are several elements of the figure that are not explained or intuitively clear. It's almost impossible to see the difference in markers between dashed and dotted here. The authors could use colours instead or make it larger. What are the black lines indicating at the right hand edges of Fig3 E and H? Is the distinction between Left and Right critical here? These figures seem to have different black lines at the right of each figure but its not clear what they indicate, or why the blue lines start high (vs red low) in both E and H.


Review 4

Summary and Contributions: In this work the authors propose to investigate the computational role of higher-order brain areas such as the PFC during decision-making tasks. The approach combines the principle of efficient coding with the formalism of MDPs. Specifically, starting off with an MDP whose states represent the trial action-observation history, the authors propose two methods for compressing this very large state space in order to preserve only task-relevant information – one for model-based behavior, the other one for habitual behavior. The authors find that, taking into account the behavior type (model-based versus habitual) and the temporal structure of the task, the compressed state space representation can explain whether the PFC shows sequential or persistent activity, as reported in perceptual decision-making paradigms with rodents and primates. The authors then describe a linear dynamical system that, under limited representational capacity, predicts well the activity of neurons at the individual and the population level. [After reviewing the authors' responses and the other reviewers' comments I believe my original score to be appropriate.]

Strengths: The computational role of the PFC is a topic of major interest in neuroscience today. There is a lot of evidence showing that this brain area has highly dynamic activity during perceptual decision-making tasks for example, but there is a lack of concrete hypotheses and models that explain why that is the case and how that activity leads to the observed behavior. This work offers an attempt at interpreting the function of these brain areas, from the point of view of state space compression, by borrowing the language of RL and MDPs specifically. There is, for this reason, an explicit connection between the neuroscience literature and its current interests and the AI and reinforcement learning fields that I find very interesting and relevant for this audience. This study is well motivated and the authors do a good job at citing the relevant literature. The tasks and neural data used in the analyses come from two prominent studies with rich data and careful task design and training. The idea that the activity of a neuron, or a population of neurons, reflects the particular state the agent is in within the a maximally compressed state space of the task is a simple yet compelling idea. And novel to the best of my knowledge. The compression methods proposed by the authors are well explained, in simple terms but rigorously and with enough detail, with clear, succinct mathematical formalism, both for model-based and habitual agents. These two methods are then applied to two behavioral paradigms involving perceptual decisions in rodents and primates. The results demonstrate well the general point the authors are trying to make here – the state transitions resulting from the compressed state space manifest two categories of dynamics observed in PFC during these tasks: persistent and sequential activity. The authors also propose a linear gaussian model that represents a more plausible setting and allows for a more direct comparison with the neural data. In this model the noise affecting the representations naturally imposes a need for representational efficiency under a limited capacity. The results using this model are useful in that they provide a more quantitative comparison with the neural population data. In fact there is a strong resemblance between the model and the neural data as measured by the demixed PC analysis. Overall the authors claims seem to me well supported by the empirical evidence provided.

Weaknesses: The main criticism I have about this work is that the results are mostly qualitative in nature. Evidence is shown suggesting that a broad categorical difference in neural activity (persistent versus sequential) can be approximated by the state representations resulting from the compression methods proposed, under different task requirements. A more quantitative comparison is done for one of the tasks, which only reflects the persistent type of activity. Even with respect to this more quantitative analysis of the neural data, the comparison is still relying on a visual resemblance between the neural data and the model results. This is particularly true for the single unit activity. It is not clear what the criteria were for selecting those two particular neurons in figure 5 (a, b) for both model and neural data. It would also be important to present a measure of how similar the individual neurons were across the population. The connection to the efficient coding hypothesis is only briefly touched on. I think this deserved to be expanded a bit. In particular, when applying these principles to areas that are not directly involved in perception and most likely involved in very flexible behavior, with evolving demands. With respect to significance and overall impact, the results in this paper, although interesting, are not groundbreaking. The impact of this paper to the community is in my view attached to providing a novel, concrete way to think about the role of higher brain areas, that has to do with efficient state representation.

Correctness: The claims and methods in this work are correct to the best of my knowledge. I did not encounter any flaws in the methodology and the claims seem to be substantiated with the empirical data provided.

Clarity: The paper is well written. The ideas are clearly exposed and I didn't find it difficult to follow the line of reasoning.

Relation to Prior Work: I believe the authors do a good job at pointing to the relevant prior work.

Reproducibility: Yes

Additional Feedback: Overall I found this paper interesting and with compelling ideas. A couple of questions and minor criticisms: - The comparison with persistent neural activity in PFC is well addressed in section 3.2. With respect to the sequential neural dynamics, there is only a coarse demonstration in section 3.1 that the compressed state space for the FDT task results in a time dependent state representation. It would have been nice to see this comparison in a more quantitative form, as in section 3.2. Is there a reason for why the data in [9] couldn't be used for the same kind of analysis? - The habitual behavior compression strategy seems to explain the neural data better. The authors justify this with the fact that the subjects are most likely overtrained in these tasks and develop habitual behavior. I empathize with this explanation. However, if I understand it correctly, this means that there isn’t a compelling application of the model-based compression strategy in this study. Perhaps it would be possible to apply the method to a different task for which subjects exhibit model-based behavior. - The connection between the state space compression procedures and the linear gaussian model wasn't clear to me. - For the experiment in section 3.2, the compression results indicate state persistence during the delay period, for the habitual condition. It also suggests the existence of different states (or neurons) dedicated to remembering different f1 frequencies. Is there any indication in the neural data that this is indeed the case, both for the animal data as well as the result of the LDS modeling – i.e. that the neurons in PFC show different preferences across the frequency space?

[Author Response · NeurIPS 2020]

We would like to thank the reviewers for the encouraging and valuable comments on our paper. We structured our response into six categories:

**1. Method criticism (R1, R3):** *R1 criticised the lack of a learning algorithm, and R3 questioned the practical utility of our approach.*

The key goal of our study is to clarify the objectives that underlie neural activities in higher-order brain areas (such as the prefrontal cortex). We do so by showing that one objective—efficient compression of task-spaces—gives rise to the type of activities measured in these areas. The questions of how to learn those representations, or how they could be utilized, are, of course, of high interest. However, they really constitute a second or even third step. Before asking them, we need to first establish a clear, mathematical objective, i.e., what exactly prefrontal areas are trying to achieve. It may be surprising, but currently no such objective exists.

**2. Conceptual novelty (R3):** *R3 questioned the conceptual novelty of our submission.*

Our main conceptual contributions are three-fold: First, we provide a clear hypothesis for the function of higher-order brain areas, which we support with a comparison to neural activity. Second, we consider both state space ('model-based') and policy ('habitual') compression. Third, we link the type of compression (state-space vs policy) to the animal's behavior. We thank R3 for the references (which we will include), but note that both Poupart et al, 2003 and Bertsekas, 1995 only consider the state-space-compression case, not the policy compression case. Indeed, policy compression is a novel concept to our knowledge, and the two compression strategies have not previously been compared.

**3. Habitual vs model-based systems (R1):** *R1 remarks that the habitual system requires a model-based system, which is deemed wasteful.*

R1 raised a very deep question: why have a separate habitual system, given that one will need a model-based system anyways. While we do not have an answer to this question, we note that this is exactly what brains seem to do: they have both habitual and model-based systems that are thought to either cooperate or compete in order to enable efficient control (for a recent discussion see e.g. Kool, Cushman & Gershman, 2018).

**4. Match to data (R1, R5):** *R1 finds that the model of the somatosensory working memory task does not match the data because it misses ramping dynamics. R5 asked for a more quantitative comparison of data and model.*

The mismatch of data and model noted by R1 must be a misunderstanding. The lower panel of Fig 5B, for example, shows a model neuron with ramping up activity towards F2. Moreover, the second, condition-independent demixed PC (Fig. 5D) also shows ramping during the delay period. Indeed, to really compare data and model, one has to compare the population data, which we here do using demixed PCA. (We would like to emphasize that we mainly show the single neuron examples because this is the standard when assessing models of these tasks, see e.g. refs [14] and [17]). We completely agree with R5, though, that one would eventually want to have a clear metric to compare model and data. However, at this point in time, most models of higher-order areas such as the PFC fail to even demonstrate a qualitative match to data. Second, and maybe for that reason (!), there is currently virtually no established protocol to compare neural data to model data for these higher-order areas. We note that these areas pose their own problems because of the flexibility and dynamics of their responses, so that establishing a metric is a non-trivial task. In the absence of such a protocol, we therefore prefer to simply visualize the match.

**5. Additional analysis:** *(a) R1 requested analysis of reaction times in the delayed licking task. (b) R5 suggested to model a model-based behaviour task. (c) R3 requested a comparison of the compressed and history state. (d) R5 suggested to model the delayed licking task with a linear system.*

(a) This is an excellent suggestion, but unfortunately, the recorded data for the delayed licking task does not allow to analyze reaction times (Inagaki et al, personal communication.) (b) Also a great suggestion, but unfortunately, there is currently no well-established model-based behavior for which neural activities have been recorded. A key problem here is that it is very hard to elucidate whether an animal is behaving habitual or model-based (see e.g. Akam et al., 2015). (c) In Figures 3,4 and 5, we have a qualitative comparison of the compressed state with the history state. Fig 5G, e.g., shows how the compressed state lies in a subspace of the history state. (d) Linear systems are too rigid to capture this task, because one action may influence the next action through the state, and because actions are in general a non-linear function of state. This is an example why in the future it will be important to look for more powerful parametrizations.

**6. Technical problems:** *R3 finds that eq. 4 is trivially true.*

In eq.4, the two distributions that are marginalized over are different: one is factored according to the history policy and the other according to the compressed policy. We understand R3's confusion and will make this dependency more explicit.

[Meta-Review · NeurIPS 2020]

This is a nice contribution in that it combines several different approaches (efficient coding, neuroscience/neural modeling, MDPs) in a conceptually novel way (R1, R4, R5), with R4 commenting that it’s likely to be of great impact to the wider community. On the other hand, R3 saw limited conceptual novelty and believes that some prior work on policy compression has been understated. In general, I’m inclined to agree with other reviewers that it’s fairly well-positioned with regard to prior work (R1). R4 praised the clarity of the writing, and other reviewers didn’t have any issues with the presentation. R5 expressed concern that the results are mainly qualitative, and not particularly novel, despite the novelty of the approach itself. One major point that came up among reviewers was the lack of a plausible method for learning. R1 argued that it’s difficult to separate the two, and I do have a concern about the applicability of their approach to more general problems requiring learning (as R1 mentions, it’s likely to be intractable). R4 didn’t consider this within the scope of the current paper, but did ask for further comment on how compression impacts further learning, and echoed R3’s concerns about how to generalize to more complex tasks. It’s not clear to me that these were adequately addressed in the rebuttal, and I think that these limitations should be discussed in the paper. Overall seems well-written, and on balance seems to provide an interesting perspective and set of results that link efficient coding with the MDP formalism, backed up by empirical neuroscientific data. Hence I am inclined to recommend accept.